# Combined Organic and Inorganic Fertilization Can Enhance Dry Direct-Seeded Rice Yield by Improving Soil Fungal Community and Structure

Xiaohong Guo [1], Jiajun Liu [1], Lingqi Xu [1], Fujing Sun [2], Yuehan Ma [1], Dawei Yin [1], Qiang Gao [1], Guiping Zheng [1] and Yandong Lv [1,*]

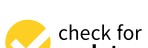



[1] Heilongjiang Provincial Key Laboratory of Modern Agricultural Cultivation and Crop Germplasm, College of Agriculture, Heilongjiang Bayi Agricultural University, Daqing 163319, China; guoxh1980@163.com (X.G.); ljj981643044@163.com (J.L.); xulingqi1994@163.com (L.X.); mayhan0517@163.com (Y.M.); dongjingcheng2002@yahoo.com.cn (D.Y.); gqiang3516@163.com (Q.G.); riceflower1960@163.com (G.Z.)

[2] Qingdao Pony Test Co., Ltd., Qingdao 266104, China; christy39@126.com

\* Correspondence: lvyd_1978@163.com; Tel.: +10-0459-6819181

**Abstract:** Direct seeding of rice has emerged as a strategy for sustainable rice (*Oryza sativa* L.) production because of advantages, such as fewer production links, labor and farmland water-saving, easy mechanization, and high economic benefits. However, few studies have investigated the effects of different organic fertilizers on soil fungal community and rice yield in dry direct-seeded paddy fields. In order to select the best combination of organic fertilizer and chemical fertilizer, field experiments were used to evaluate the role of no fertilizer (F0); CF, conventional NPK fertilizer, OF1, biochar + conventional NPK fertilizer; OF2, seaweed bioorganic fertilizer + conventional NPK fertilizer; OF3, Jishiwang bioorganic fertilizer + conventional NPK fertilizer; and OF4, attapulgite organic fertilizer + conventional NPK fertilizer on microbial structure and diversity and rice yield. Under Jishiwang bioorganic fertilizer + conventional NPK fertilization, the number of fungal OTUs was 365 and ranged from 1 to 9. The Ascomycota relative abundance was increased by 28.25% under Jishiwang bioorganic fertilizer application compared with CF, but the Basidiomycota decreased. Sordariomycetes and Leotiomycetes relative abundances were increased under organic fertilization. The relative abundance of dung saprotrophs, fungal parasites, and leaf saprotrophs was increased under organic fertilizer compared to CF, and animal pathogens decreased, but organic fertilizers also increased plant pathogens. Rice yield was increased under Jishiwang bioorganic fertilizer + conventional NPK fertilizer and was positively correlated with Ascomycota and Sordariomycetes relative abundances. The use of Jishiwang bioorganic fertilizer + conventional NPK fertilizer improves fungal community diversity and rice yield.

**Keywords:** yield and yield components; organic-inorganic fertilizers; dry direct-seeded rice; high-throughput sequencing; fungal diversity; ecological function

## 1. Introduction

Rice (*Oryza sativa* L.) is one of the most important food crops in the world, which can meet the food needs of more than 3 billion people in more than 100 countries across the world [1]. However, due to the increasing shortage of freshwater for crop irrigation and the reduction of the labor force in the rice-planting industry, traditional Chinese rice cultivation is under threat [2,3]. In this context, dry direct-seeded rice has emerged as a simplified and sustainable cultivation technology of great significance, but yields may be limited [4]. Due to phenological and physiological differences [5], the traditional fertilization technology applied to transplanted rice is not suitable for dry direct-seeded rice [6]. Till now, most studies have focused on the effect of chemical fertilizers on dry direct-seeded rice [7,8]. The

excessive use of chemical fertilizers is the main cause of resource waste and environmental damage [9]. Organic fertilizers contain multiple nutrients required for plant growth, such as trace elements, sugars, and fats. They can improve the physicochemical properties of the soil, enhance microbial abundance and activities, and improve crop yield and quality [10,11]. Therefore, it is important to identify the best combination of organic and chemical fertilizers with respect to dry direct-seeded rice.

The soil microbial community includes two important components, soil fungi and bacteria. The soil fungal community plays diverse functions in the ecosystem and is involved in many ecological processes, such as improving plant nutrient utilization through biological nitrogen fixation and phosphorus dissolution in the soil and rhizosphere and enhancing plant growth and development [12–14], thereby playing an important role in various types of terrestrial ecosystems. Fungi are more tolerant to water stress compared with bacteria [15] because hyphae can explore the soil more profusely, gaining greater access to soil water [16]. Better access to soil pores through fungal hyphae allows a more efficient nutrient acquisition [17], thus alleviating the pressure of water stress. On the other hand, soil fungi also have important effects on soil organic compounds [18], thus affecting soil nutrient conversion [19]. Some studies also show that soil fungi are more sensitive to fertilization management measures than bacteria [20]. It has been reported that the long-term application of inorganic fertilizers reduces fungal diversity in soil under wheat cultivation [21]. Compared with chemical fertilizers, organic fertilizers improve soil fungal diversity and community structure [22]. Likewise, the combination of organic and inorganic fertilizers may increase the diversity of soil bacteria and fungi, making its distribution more homogeneous than that found in soils under inorganic fertilizers and in unfertilized soils [23]. It has also been reported that even in the short term, replacing part of the chemical fertilizer with organic fertilizers can optimize soil fungal community structure, promote soil beneficial fungi multiplication, and inhibit harmful soil bacteria [24]. Therefore, exploring the changes in soil fungal communities under different fertilization regimes is important to improve our understanding of the rice-microbial ecosystem.

At present, studies on the impact of organic fertilizers on soil fungal diversity are limited [25,26], and the consequences of such practices in dry direct-seeded rice fields were rarely reported. In this study, Illumina Miseq high-throughput sequencing was used to analyze the influence of different organic fertilizers on the soil fungal community and rice yield in a dry direct-seeded rice field, and to explore the best combination of organic and chemical fertilizers, in order to provide a theoretical basis for optimizing the soil microflora and selecting appropriate types of organic fertilizer in dry direct-seeded rice field.

## 2. Materials and Methods

### 2.1. Experimental Site

The fertilization regimes described in the following section were applied for three consecutive dry direct-seeded rice campaigns (2018–2020) under field conditions; the plots were located in Youyi farm, Heilongjiang Province (46.45° N, 131.49° E). The climate in this region corresponds to middle temperate continental monsoon, with annual mean temperature and precipitation of 3.1 °C and 501.2 mm, respectively (Figure 1). The soil type in this area is meadow black soil. The physicochemical properties of the soil at a depth of 0–20 cm are shown in Table 1.

### 2.2. Experimental Design

Dry direct-seeded rice was subjected to 6 experimental treatments under a complete randomized design, with the fertilization regime as the sole variable factor, including no fertilizer (F0), conventional NPK fertilizer (CF), and 4 combined organic-inorganic fertilizers: biochar (Shenyang Longtai Biological Engineering Co., Ltd., Shenyang, Liaoning Province, China) + conventional NPK fertilizer (OF1), seaweed bioorganic fertilizer (Qingdao Pulebaiwo Biotechnology Co., Ltd., Qingdao, Shandong Province, China) + conventional NPK fertilizer (OF2), Jishiwang bioorganic fertilizer (Jiamusi Sanxing Agricultural Technical Ser-

vice Co., Ltd., Jiamusi, Heilongjiang Province, China) + conventional NPK fertilizer (OF3), and attapulgite organic fertilizer (Dingfengyuan concave High-tech Development Co., Ltd., Zhangye, Gansu Province, China) + conventional NPK fertilizer (OF4). Conventional NPK fertilizer (CF) was used as the control, and no fertilizer use (F0) was also included.

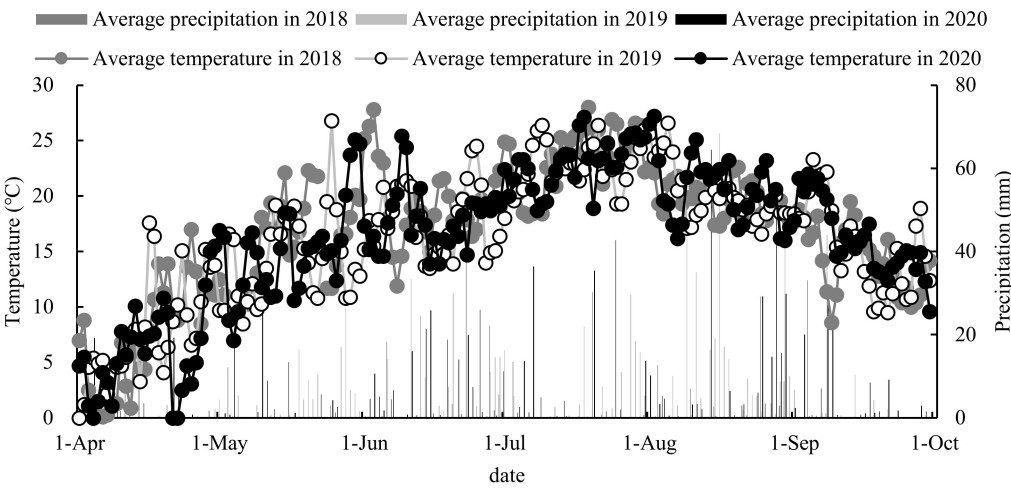

**Figure 1.** Temperature and rainfall during dry direct-seeded rice cultivation during 2018–2020.

**Table 1.** Physiochemical properties of the soil at the experiment site between 2018 and 2020.

| Year | Hydrolyzable Nitrogen (mg kg$^{-1}$) | Available Phosphorus (mg kg$^{-1}$) | Available Potassium (mg kg$^{-1}$) | Organic Matter (g kg$^{-1}$) | pH |
|------|------|------|------|------|------|
| 2018 | 106.67 | 29.55 | 171.95 | 27.70 | 5.91 |
| 2019 | 137.40 | 24.40 | 159.90 | 39.00 | 6.70 |
| 2020 | 128.46 | 25.38 | 163.43 | 40.00 | 6.32 |

In the conventional fertilization regime, a total of 204.05 kg N ha$^{-1}$ (urea, 46% N) was applied, distributed as basal fertilizer (88.82 kg N ha$^{-1}$), tillering fertilizer (69.00 kg ha$^{-1}$), and panicle fertilizer 46.23 kg N ha$^{-1}$. The total amount of phosphate fertilizer (calcium triple superphosphate, 46% $P_2O_5$) was $P_2O_5$ 80.04 kg ha$^{-1}$, all as base fertilizer, and the total amount of potassium fertilizer (potassium sulfate, 50% $K_2O$) was $K_2O$ 115.00 kg ha$^{-1}$, divided into basal (63.75 kg $K_2O$ ha$^{-1}$) and panicle (51.25 kg $K_2O$ ha$^{-1}$) fertilizers. Organic fertilizers comprised biochar (organic matter 26.5%); seaweed bioorganic fertilizer (N + $P_2O_5$ + $K_2O$ ≥ 6.0%, organic matter ≥ 50.0%, living bacteria ≥ 20 million g$^{-1}$); Jishiwang bioorganic fertilizer (N + $P_2O_5$ + $K_2O$ ≥ 6.0%, organic matter content ≥ 40.0%, living bacteria ≥ 20 million g$^{-1}$), and attapulgite organic fertilizer (N + $P_2O_5$ + $K_2O$ ≥ 6.0%, organic matter ≥ 41.5%); these four compounds were applied before tillage at the doses of 15 t ha$^{-1}$, 300 kg ha$^{-1}$, 300 kg ha$^{-1}$, and 750 kg ha$^{-1}$, respectively.

The rice variety Longjing 31 (11 leaves on the main stem) was used in this experiment; this genotype is characterized by about 130 days to maturity and an active accumulated temperature ≥ 10 °C of about 2350 °C.

Before sowing, the seeds were coated with the Liangdun seed coating agent, and the dry seeds were mechanically sown on 30 April 2018, 18 April 2019, and 19 April 2020. In all cases, the seeding rate was 210 kg ha$^{-1}$, and the seeding depth was 2 cm, with a row spacing of 20 cm. Each experimental plot covered 300 m$^2$ (length 20 m, width 15 m), and 3 replicates per treatment were set, making a total of 18 plots. Basal fertilizer was applied at 4 cm aside from the seedling belt, 5 cm depth; a soil covering chain was used, and the soil was compacted twice after sowing. Harvest was performed on 20 September, 23 September, and 27 September in the rice production cycles of 2018, 2019, and 2020, respectively.

### 2.3. Soil Sampling and Rice Yield and Yield Components Measurements

Four points in each experimental plot were randomly selected to obtain four soil samples from the 0–10 and 10–20 cm soil layers after the rice harvest in September 2020; subsamples from each layer were collected and mixed (same layer). Fresh soil samples were transported in ice bags to the laboratory and kept at −80 °C until analysis. In addition, 3 points in each plot were randomly selected at rice maturity to collect 30 representative panicles to calculate the theoretical yield based on the measurements of yield components. The number of panicles per hill was determined and converted to the number of panicles per square meter. Filled and unfilled spikelets were separated by submerging them in tap water, counted, and weighed. Spikelet number per panicle, seed-setting rate, and 1000-grain weight were determined. The actual yield was assessed by collecting rice plants from a 5 m$^2$-area in each plot; final data were obtained after adjusting the moisture content to 14%.

### 2.4. Soil DNA Extraction and Illumina HiSeq Sequencing

Total fungal DNA was extracted from the soil samples collected using Power Soil DNA Isolation Kit (MO BIO Laboratories), according to the manufacturer's protocol. DNA quality and quantity were assessed by measuring the absorbances and calculating 260 nm/280 nm and 260 nm/230 nm ratios. The DNA was stored at −80 °C until further processing.

The fungal ITS1 region was amplified using primers ITS1-F (5′-CTTGGTCATTTAGAG GAAGTAA-3′) and ITS2-R (5′-GCTGCGTTCTTCATCGATGC-3′) combined with Illumina adapter sequences and barcodes. The PCR was performed in a total reaction volume of 20 μL: H$_2$O 13.25 μL, 10× PCR ExTaq Buffer 2.0 μL, DNA template (100 ng/mL) 0.5 μL, primer 1 (10 mmol/L) 1.0 μL, primer 2 (10 mmol/L) 1.0 μL, dNTP 2.0 μL, and ExTaq polymerase (5 U/mL) 0.25 μL. After an initial denaturation at 95 °C for 5 min, amplification was achieved by 30 cycles of 30 s at 95 °C, 20 s at 58 °C, and 6 s at 72 °C, followed by a final extension step at 72 °C for 7 min. Then, the amplified products were purified and recovered using 1.0% agarose gel electrophoresis. The library construction and sequencing steps were performed by Beijing Biomarker Technologies Co., Ltd.

### 2.5. Bioinformatics and Statistical Analyses

The bioinformatic analysis in this study was completed using the Biomarker Biocloud Platform (www.biocloud.org, accessed on 5 May 2021). To obtain the raw tags, paired-end reads were merged using FLASH (v1.2.7, http://ccb.jhu.edu/software/FLASH/, accessed on 16 May 2021) [27]. Then, raw tags were filtered and clustered in the next steps. The merged tags were compared with the primers, and the tags with more than six mismatches were discarded by the FASTX-Toolkit. Tags with an average quality score < 20 in a 50 bp sliding window were truncated using Trimmomatic (http://www.usadellab.org/cms/?page=trimmomatic, accessed on 25 May 2021) [28], and tags shorter than 350 bp were removed. We identified possible chimeras by employing UCHIME, a tool included in mothur (http://drive5.com/uchime, accessed on 28 May 2021). The denoised sequences were clustered using USEARCH (version 10.0), and tags with a similarity ≥ 97% were regarded as an operational taxonomic unit. Taxonomic classification was assigned to all units by searching against the Silva database (http://www.arb-silva.de, accessed on 8 June 2021) using the UCLUST classifier in QIIME.

Four fungal alpha diversity indices (Chao1 richness estimator, ACE richness estimator, Shannon–Wiener diversity index, and Simpson diversity index) were calculated by mothur v1.30 [29]. The variation in fungal communities among soil samples was visualized by a principal coordinates analysis (PCoA) based on the Bray–Curtis dissimilarity index using the Vegan package.

The data from the 3 years were pooled, and group means were assessed using one-way ANOVA followed by the least significant difference (LSD) test, performed using IBM SPSS v20.0 statistical software (SPSS, Chicago, IL, USA). A Pearson correlation analysis was

conducted to assess the relationships between yield and dominant soil fungi at phylum and class levels. Figures were generated using GraphPad Prism 7 and Origin 2021 software.

## 3. Results

### 3.1. Paddy Soils Sequencing Throughput

The raw sequences obtained through Illumina sequencing from the soil samples collected were qualitatively controlled, assembled, and filtered for chimeras. A total of 49,855–62,521 raw reads and 46,126–58,375 effective reads were obtained for fungi (Table 2); the effective reads accounted for 89.80–93.37% of the raw reads. The number of raw reads and effective reads and the percentage of effective reads were lower under all combined treatments (OF1, OF2, OF3, and OF4) than CF. The sequences were similar in length across all treatments, ranging between 241 and 251 bp, and the coverage level was over 0.99 in all cases, indicating that the sequencing data were reliable and could truly reflect the community composition of soil fungi.

**Table 2.** Sequencing throughput data for soil fungi according to fertilization treatments.

| Treatment [1] | Raw Reads | Effective Reads | Average Sequence Length | Effectiveness (%) | Coverage |
|---|---|---|---|---|---|
| F0 | 49,855 | 46,126 | 224 | 92.50 | 0.9989 |
| CF | 62,521 | 58,375 | 241 | 93.37 | 0.9991 |
| OF1 | 55,992 | 50,557 | 251 | 90.28 | 0.9990 |
| OF2 | 56,730 | 50,947 | 248 | 89.80 | 0.9990 |
| OF3 | 57,700 | 52,900 | 247 | 91.66 | 0.9991 |
| OF4 | 58,259 | 52,598 | 245 | 90.28 | 0.9990 |

[1] F0, no fertilizer (F0); CF, conventional NPK fertilizer, OF1, biochar + conventional NPK fertilizer; OF2, seaweed bioorganic fertilizer + conventional NPK fertilizer; OF3, Jishiwang bioorganic fertilizer + conventional NPK fertilizer, OF, attapulgite organic fertilizer + conventional NPK fertilizer.

### 3.2. Effect of Organic-Inorganic Fertilizers on the Alpha Diversity of the Soil Fungal Community

Figure 2a–d shows the effect of the fertilization treatments applied to the dry direct-seeded rice on several fungi alpha diversity indices. Unfertilized plots (F0) exhibited smaller fungi species richness (ACE and Chao1) and diversity (Simpson) than fertilized plots. Moreover, species richness under combined organic-inorganic fertilizers tended to diminish compared with CF, but no significant differences were detected. Among fertilized treatments, OF1 was associated with the lowest ACE and Chao1 indices and the highest Simpson and Shannon indices, followed by those found for OF3. OF2 and OF4 led to minor changes in these parameters compared with CF. Briefly, treatment with organic fertilizers had different effects. The ACE, Chao1, and Simpson and Shannon indexes changed between different treatments, but there was no significant difference with CF. This shows that organic fertilizer type and composition are relevant for determining the alpha diversity of soil fungi communities in dry direct-seeded rice fields.

### 3.3. Effect of Combined Organic-Inorganic Fertilizers on Soil Fungal OTUs

As the Venn graph depicts, the number of fungal OTUs shared by all soil samples was 365 (Figure 3), while a reduced number of unique OTUs were recognized for treatments F0, CF, OF1, OF2, OF3, and OF4: 3, 7, 9, 1, 2, and 6, respectively. This result indicates that in the dry direct-seeded rice field, the application of biochar tended to increase the number of fungal species in the soil compared with conventional fertilization. In contrast, the application of the seaweed bioorganic fertilizer and the Jishiwang bioorganic fertilizer reduced these values, a finding that may be related to the competitive pressure exerted by the microorganisms contained in both organic fertilizers.

The differences in soil fungal community composition among soils subjected to different fertilization regimes during dry direct-seeded rice cultivation were subsequently analyzed by a principal coordinates analysis (PCoA) at the OTU level (Figure 4). This analysis revealed that fungal community variability under each fertilization treatment was

explained by 37.69% and 11.74% in the axes PC1 and PC2, respectively; the total interpretation rate was 49.44%. It may be noticed that OTUs associated with the fungal community in non-fertilized plots (F0) were mainly distributed on the zero axis of PC2, scattered and away from those corresponding to fertilized treatments. OTUs representing the fungal community in CF, OF1, and OF3 were concentrated in the second quadrant and clustered, suggesting that the fungal community composition in these samples was similar. OTUs linked to the fungal community in OF2 and OF4 occupied mainly the third quadrant, close to each other and CF, OF1, and OF3, indicating that the application of organic-inorganic fertilizers influenced the fungal community structure in dry direct-seeded rice fields.

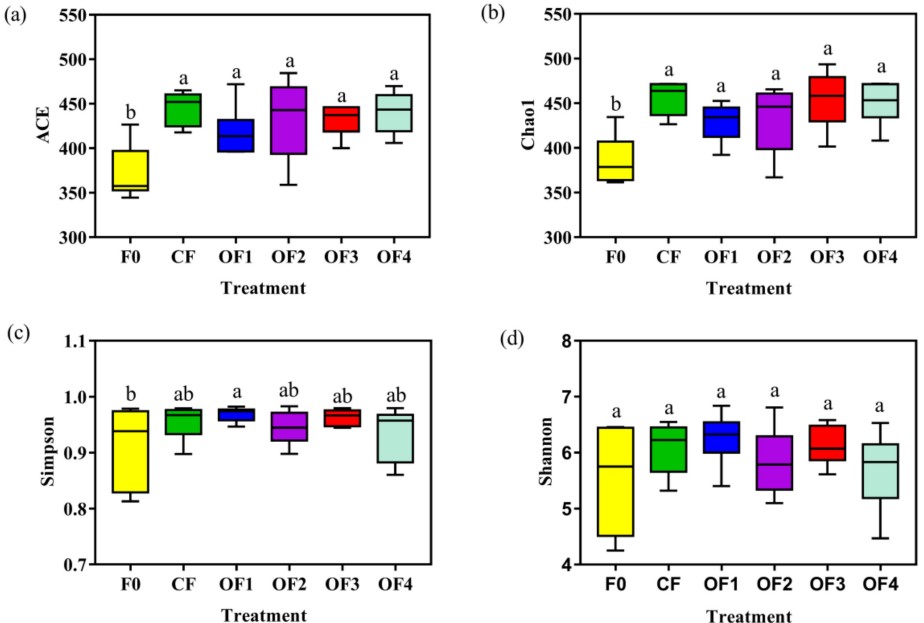

**Figure 2.** Alpha diversity (**a**, ACE index; **b**, Chao1 index; **c**, Simpson index; and **d**, Shannon index) of soil fungal community. F0, no fertilizer (F0); CF, conventional NPK fertilizer, OF1, biochar + conventional NPK fertilizer; OF2, seaweed bioorganic fertilizer + conventional NPK fertilizer; OF3, Jishiwang bioorganic fertilizer + conventional NPK fertilizer, OF4, attapulgite organic fertilizer + conventional NPK fertilizer. The same letters above the bars are not significantly different among treatments (ANOVA, LSD test, $p < 0.05$).

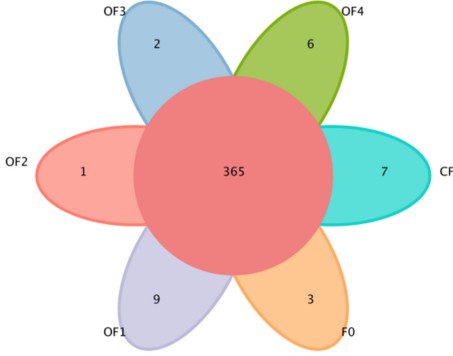

**Figure 3.** Venn graph showing the number of shared and unique fungal OTUs among fertilization treatments. F0, no fertilizer (F0); CF, conventional NPK fertilizer, OF1, biochar + conventional NPK fertilizer; OF2, seaweed bioorganic fertilizer + conventional NPK fertilizer; OF3, Jishiwang bioorganic fertilizer + conventional NPK fertilizer, OF4, attapulgite organic fertilizer + conventional NPK fertilizer.

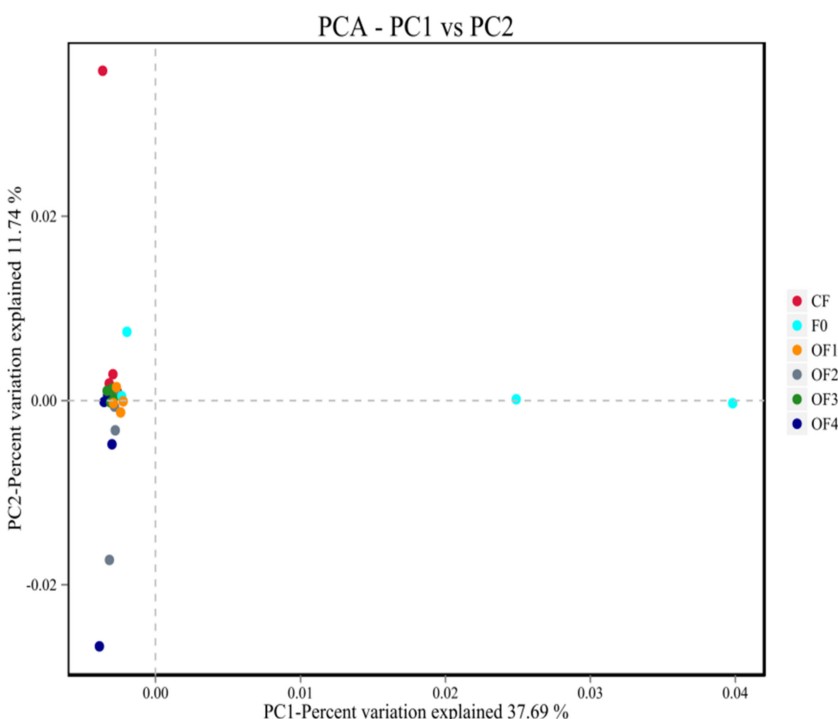

**Figure 4.** Principal coordinates analysis of soil fungal OTUs. F0, no fertilizer (F0); CF, conventional NPK fertilizer, OF1, biochar + conventional NPK fertilizer; OF2, seaweed bioorganic fertilizer + conventional NPK fertilizer; OF3, Jishiwang bioorganic fertilizer + conventional NPK fertilizer, OF4, attapulgite organic fertilizer + conventional NPK fertilizer.

### 3.4. Effect of Organic-Inorganic Fertilizers on Soil Fungal Community at Phyla and Class Levels

In this study, eight fungal phyla were identified in the soil samples analyzed, and five phyla had a relative abundance > 1%: Ascomycota, Basidiomycota, Mortierellomycota, Rozellomycota, and Chytridiomycota (Figure 5a). Compared with CF, the relative abundance of Ascomycota in OF1 and OF3-treated soils increased by 2.49% and 28.25%, respectively, while it decreased by 20.05% and 17.99% in OF2 and OF4. Interestingly, Basidiomycota's relative abundance showed the opposite trend of Ascomycota, with lower values under OF1 and OF3 compared with CF and higher values under OF2 (39.06%) and OF4 (63.51%). It is worth mentioning that the introduction of organic fertilizers in dry direct-seeded paddy soil resulted in an inhibitory effect on Mortierellomycota, reducing its relative abundance by 15.24% to 39.98% compared with CF. Rozellomycota's relative abundance under OF1, OF2, and OF4 was greater than that found in CF; the highest increase corresponded to OF2 (58.21% over CF), while the highest decrease found corresponded to OF3 (69.49% below CF). Compared with CF, the relative abundance of soil Chytridiomycota increased by 104.23% and 31.54% in rice plots subjected to OF1 and OF4, respectively, and decreased by 6.15% and 24.23% in those under OF2 and OF3. Rice plots treated with OF1 had a notably greater proportion of Ascomycota, Rozellomycota, and Chytridiomycota (compared to CF), whereas Basidiomycota was dominant in plots treated with OF2 and OF4. Mortierellomycota only occupied a dominant position in CF.

At the class level, eight dominant fungal classes with relative abundance > 1% were identified, comprising, in decreasing order, Sordariomycetes (25.00–32.77%), Mortierellomycetes (7.63–12.24%), Dothideomycetes (5.09–10.43%), Leotiomycetes (2.45–13.75%), Tremellomycetes (1.76–6.44%), Agaricomycetes (1.53–4.89%), Eurotiomycetes (1.24–5.21%), and Pezizomycetes (0.84–3.92%) (Figure 5b). The relative abundance of Sordariomycetes under OF3 and OF4 increased compared to CF (7.09% and 3.86%, respectively), while it decreased in OF1 and OF2 treatments (8.14% and 19.61%, respectively). Furthermore, the incorporation of organic fertilizers considerably decreased the relative abundance of

Mortierellomycetes and Tremellomycetes, with the lowest values for the former in OF1 (37.66% below CF) and for the latter in OF2 (72.67% below CF).

(a)

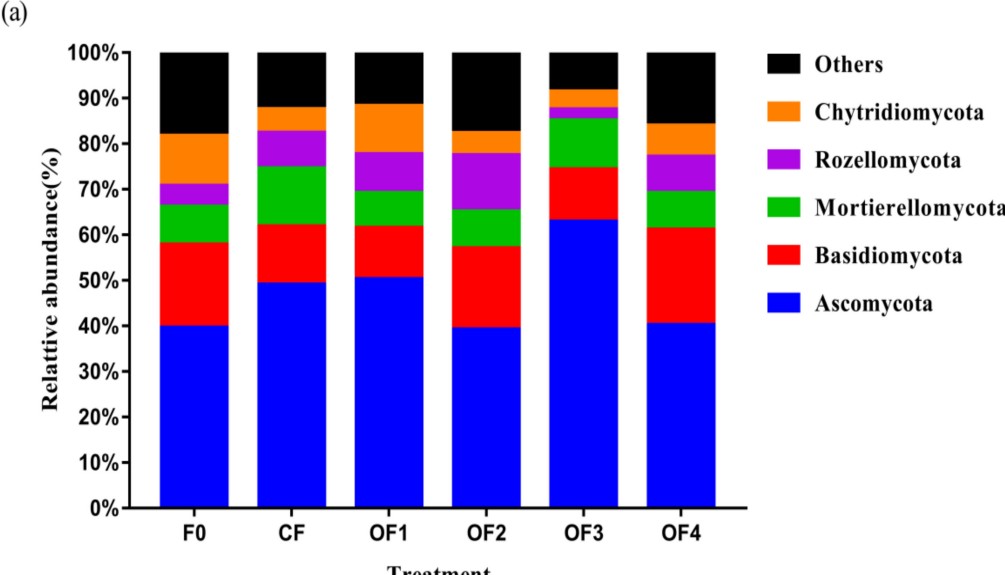

(b)

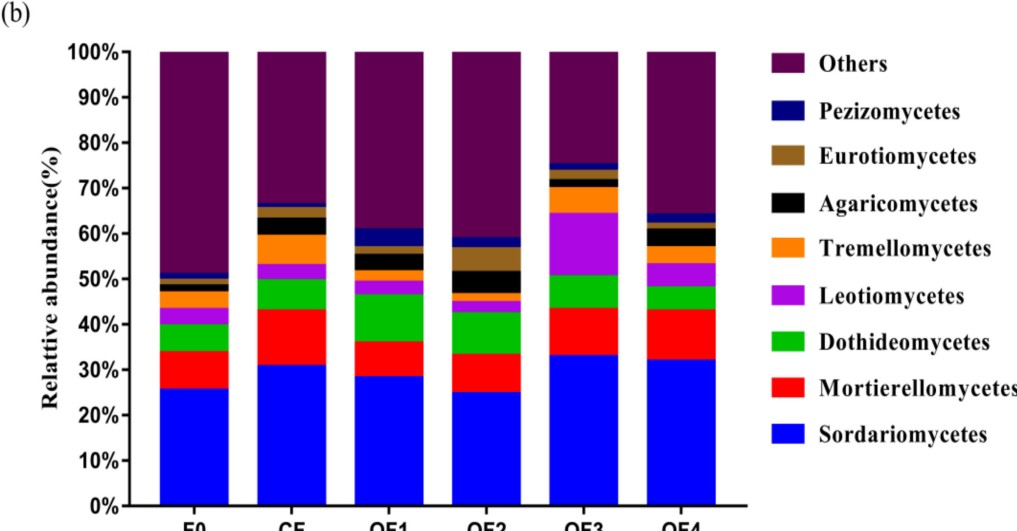

**Figure 5.** Dominant fungal phyla (**a**) and classes (**b**). F0, no fertilizer (F0); CF, conventional NPK fertilizer, OF1, biochar + conventional NPK fertilizer; OF2, seaweed bioorganic fertilizer + conventional NPK fertilizer; OF3, Jishiwang bioorganic fertilizer + conventional NPK fertilizer, OF4, attapulgite organic fertilizer + conventional NPK fertilizer.

All organic-inorganic fertilizers increased the relative abundance of Pezizomycetes; increases reached 3.67 times (OF1), 1.60 times (OF2), 0.65 times (OF3), and 1.40 times (OF4) compared to CF. The relative abundance of Dothideomycetes in the soils sampled from OF1, OF2, and OF3 showed an increasing trend compared with CF, with the greatest change in OF1. The highest relative abundance of Leotiomycetes was detected under OF3, followed by that found under OF4 (3.15 and 0.83 times over CF, respectively). The relative abundance of Agaricomycetes and Eurotiomycetes was lowest in OF3. Sordariomycetes, Pezizomycetes, Dothideomycetes, Leotiomycetes, Agaricomycetes, and Eurotiomycetes occupied a relatively dominant position in the plots exposed to the combined organic and inorganic fertilization treatments, especially OF3.

### 3.5. Ecological Function of Soil Fungi

We explored the putative ecological function of the fungal community deduced from the sequencing analysis based on the FUNGuild software (Figure 6). The soil fungal community across the 6 treatments tested could be ascribed to 10 ecological functional groups: undefined saprotroph, dung saprotroph, wood saprotroph, animal pathogen, plant pathogen, plant saprotroph, fungal parasite, soil saprotroph, endophyte, and leaf saprotroph. Notably, the relative abundance of animal pathogens was greater than that of dung saprotrophs under CF, while the relative abundance of wood saprotrophs was greater than that of dung saprotrophs under OF3 treatment, which suggests that OF3 treatment had a particular impact on functional group distribution in the fungal community.

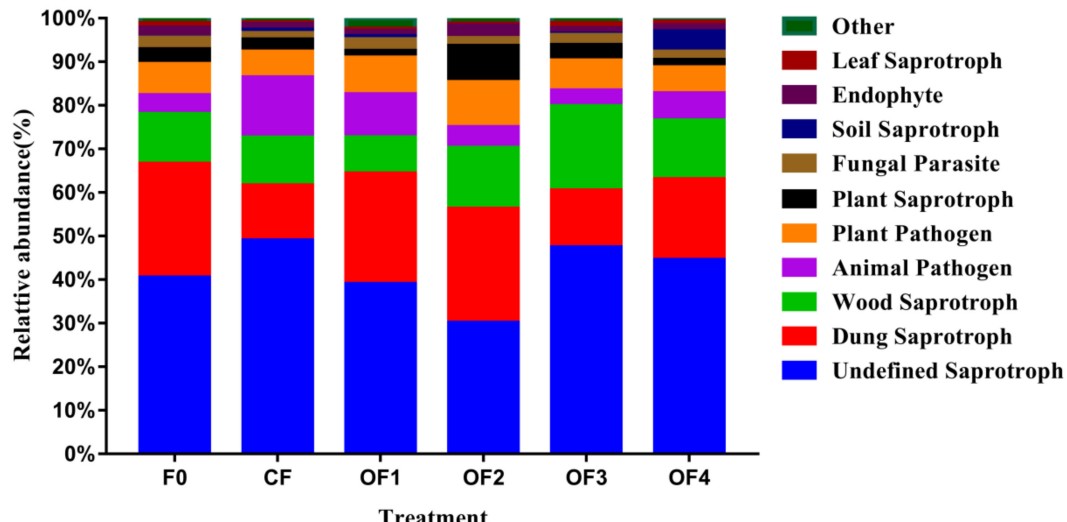

**Figure 6.** Dominant ecological functional groups in the fungal community. F0, no fertilizer (F0); CF, conventional NPK fertilizer, OF1, biochar + conventional NPK fertilizer; OF2, seaweed bioorganic fertilizer + conventional NPK fertilizer; OF3, Jishiwang bioorganic fertilizer + conventional NPK fertilizer, OF4, attapulgite organic fertilizer + conventional NPK fertilizer.

The relative abundances of dung saprotroph, plant pathogen, fungal parasite, and leaf saprotroph groups under organic-inorganic fertilizers were all higher than those found under conventional fertilizer treatment, while animal pathogens showed the opposite trend. The relative abundance of wood saprotroph and endophyte groups increased in OF2, OF3, and OF4 compared with conventional fertilization. Soil saprotrophs were at the highest relative abundance in the soil under attapulgite fertilizer. It can be seen that the long-term use of organic fertilizer increased the relative abundance of wood saprotroph and endophyte groups but decreased the relative abundance of animal pathogens.

### 3.6. Yield and Yield Components under Different Fertilizer Treatments in Dry Direct-Seeded Rice

As shown in Table 3, the number of panicles per square meter (PPSM) under OF2 significantly exceeded that of CF, reaching the highest value (668.3 panicles $m^{-2}$); OF3 and OF4 treatments also resulted in higher PPSM than CF, while OF1 resulted in a lower value. The seed-setting rate (SSR) and 1000-grain weight (1000 GW) were highest under OF4 and OF1, respectively. Both the theoretical yield and the actual yield presented the following decreasing order, OF3 > OF2 > OF4 > CF > OF1 > F0. Significant differences for OF2 and OF3 compared to CF were detected for the theoretical yield, which increased by 19.0% and 23.4%, respectively, while there was no significant difference in the actual yield among fertilization treatments. Among the treatments, the theoretical yield and actual yield associated with OF3 were the highest; this was attributed to the increase in panicle number.

**Table 3.** Yield and yield components of dry direct-seeded rice [1].

| Treatment [2] | PPSM | GNPP | SSR (%) | 1000 GW(g) | TY (Mg ha$^{-1}$) | AY (Mg ha$^{-1}$) |
|---|---|---|---|---|---|---|
| F0 | 436.3d | 44.0 c | 74.7 b | 20.88 b | 2992 d | 1924 b |
| CF | 590.0bc | 73.9 ab | 89.4 a | 21.37 ab | 8304 c | 7372 a |
| OF1 | 557.5c | 71.8 ab | 88.4 a | 22.87 a | 8075 c | 7334 a |
| OF2 | 668.3a | 74.4 ab | 92.0 a | 21.61 ab | 9883 ab | 7763 a |
| OF3 | 663.8a | 78.9 a | 88.5 a | 22.12 ab | 10,244 a | 8512 a |
| OF4 | 646.3ab | 70.6 b | 99.1 a | 21.63 ab | 8890 bc | 7570 a |

[1] PPSM, panicles per square meter; GNPP, grain number per panicle; SSR, seed setting rate;1000GW,1000-grain weight; TY, theoretical yield; AY, actual yield. [2] F0, no fertilizer (F0); CF, conventional NPK fertilizer, OF1, biochar + conventional NPK fertilizer; OF2, seaweed bioorganic fertilizer + conventional NPK fertilizer; OF3, Jishiwang bioorganic fertilizer + conventional NPK fertilizer, OF, attapulgite organic fertilizer + conventional NPK fertilizer; Different small letters in the same column indicate significant differences among treatments ($p < 0.05$).

### 3.7. Yield and Dominant Fungal Taxa in Paddy Soil: Correlation Analysis

A correlation analysis was carried out between the actual yield and the dominant fungi at the phyla and class levels to further explore the relationship between these relevant parameters (Figure 7). Dry direct-seeded rice yield was significantly positively correlated with Ascomycota relative abundance (correlation coefficient, 0.42) and significantly negatively correlated with Chytridiomycota relative abundance (correlation coefficient: −0.49), suggesting that these conditions contribute to improving rice yield. It can also be seen in Figure 7a that Ascomycota relative abundance was significantly negatively associated with that of Basidiomycota and Mortierella, and Basidiomycota, Mortierellomycota, and Chytridiomycota relative abundances also showed negative correlations with each other. These findings suggest that Ascomycota is the fungal phylum with the greatest influence on yield, and complex, competitive interactions between fungal phyla occur in the soil.

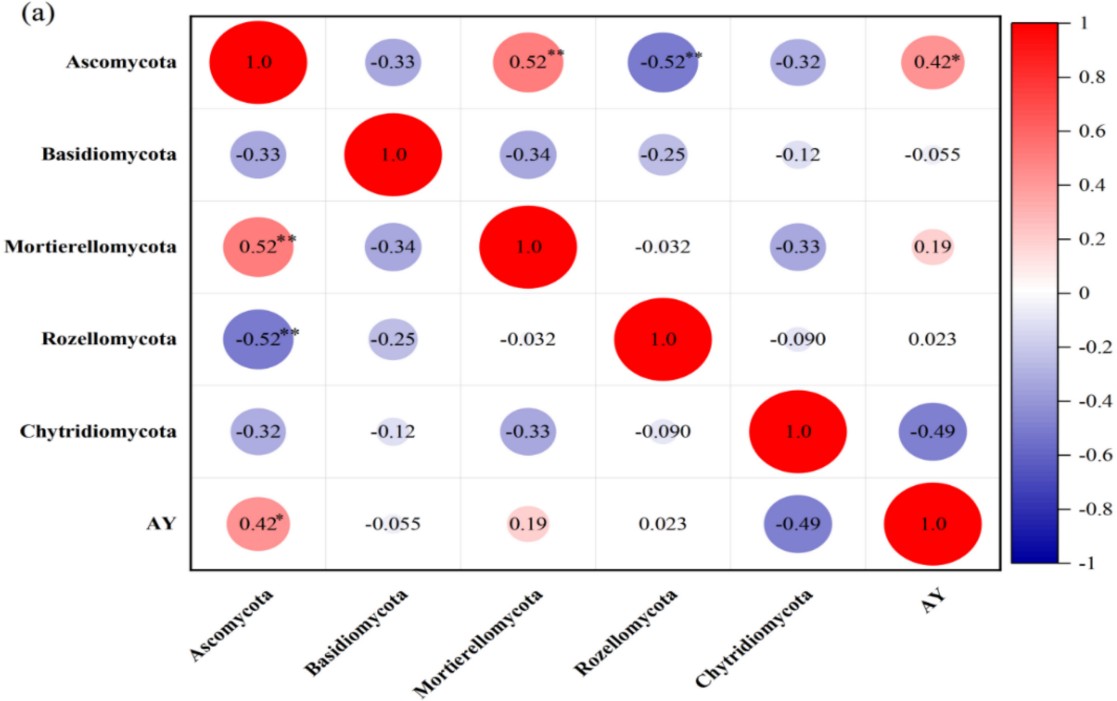

**Figure 7.** *Cont.*

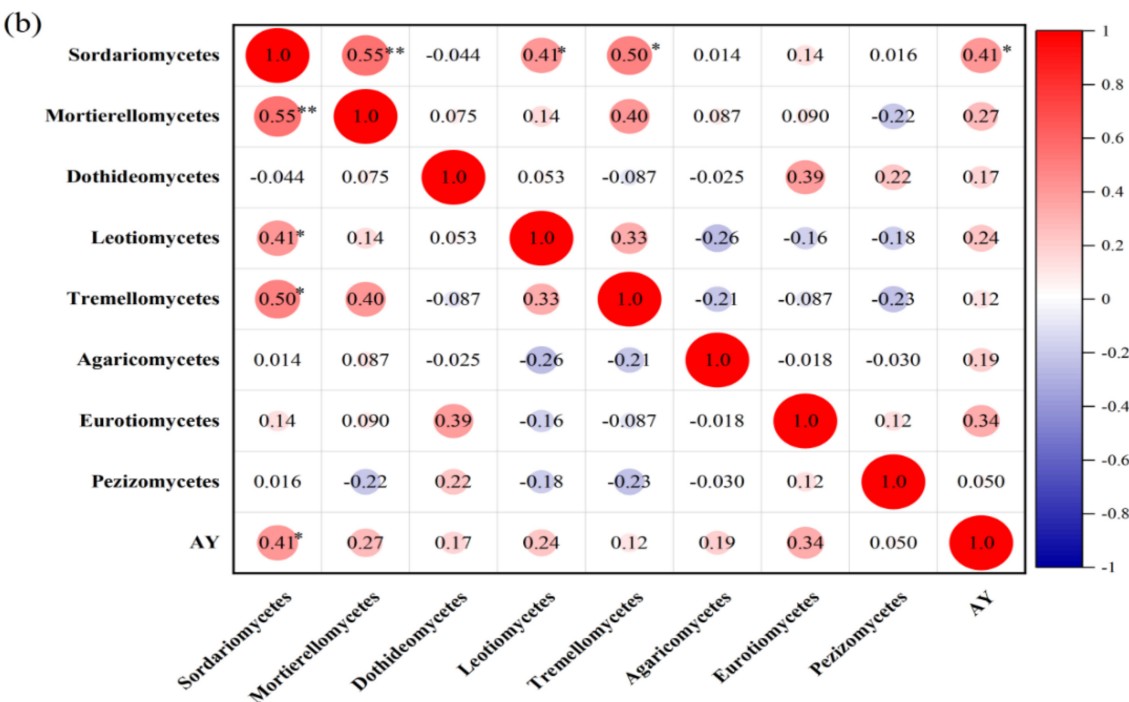

**Figure 7.** Correlation analysis of yield and dominant soil fungi at phylum (**a**) and class (**b**) levels. *, ** Significant at $P < 0.05$ and $P < 0.01$, respectively.

Figure 7b shows that at the class level, only the relative abundance of Sordariomycetes was significantly and positively correlated with dry direct-seeded rice yield, with a correlation coefficient of 0.41; the remaining dominant fungi were positively correlated with rice yield but at non-significant levels. The relationships among dominant fungi in the soil at the class level differed from those detected at the phylum level: the relationships in the former tended to be mutually beneficial–competitive. Very significant or significant positive correlations were detected between Sordariomycetes and Mortierellomycetes, Leotiomycetes, and Tremellomycetes. Dothideomycetes were negatively correlated with Sordariomycetes, Tremellomycetes, and Agaricomycetes. Agaricomycetes, Eurotiomycetes, and Pezizomycetes were also negatively correlated with each other.

## 4. Discussion

### 4.1. Effect of Organic-Inorganic Fertilizers on Dry Direct-Seeded Paddy Soil Fungal Diversity

It was reported that organic fertilizers could regulate soil microbial community structure, and through the changes in the soil carbon utilization pattern they induce, organic fertilizers may increase soil microbial functional diversity and improve soil quality [30,31]. However, it was also shown that combined inorganic and organic fertilizers weakly improved soil fungal diversity [32].

Soil fungal ACE and Chao indices under organic-inorganic fertilizer treatments showed a slightly decreasing trend compared with CF, without significant differences. These findings suggest that combined fertilization schemes in dry direct-seeded rice fields might not be conducive to enhancing soil fungal richness. Ji et al. [33] showed that the abundance of fungal species in plots fertilized with organic fertilizers was lower than in plots exposed to inorganic fertilizers, considering the same soil layer. Chen et al. [34] showed that the application of organic fertilizers significantly reduced fungal diversity, while inorganic fertilizers' addition reduced fungal richness. However, a study by Luo et al. [35] showed that both inorganic and organic fertilizers increased the taxonomic abundance of soil fungi, but the effect of the organic fertilizer was more obvious.

The findings of Hu et al. [36] highlighted that geographic isolation and fertilization systems adopted under agricultural management were important factors influencing the changes in fungal communities in response to contrasting fertilization regimes. Differences in soil type and analytical methods, among others, may account for variable results [34,37]. According to the comparative analysis of Simpson and Shannon indices, biochar application (OF1) led to the highest fungal diversity, followed by the Jishiwang bioorganic fertilizer (OF3). Li et al. [38] showed that biochar application increased fungal diversity in purple soil. It was proposed that biochar has many pores, which could provide shelter for soil microorganisms; on the other hand, pores might improve soil aeration, moisture, and nutrient availability [39,40]. However, the phenols and polyphenols, microelements, organic pollutants, and persistent free radicals present in biochar might also have negative effects on microorganisms [41–43].

We found that the application of seaweed bioorganic fertilizer (OF2) and attapulgite organic fertilizer (OF4) resulted in lower fungal diversity than that observed under conventional fertilization, indicating that the properties of the organic fertilizer itself might be relevant in terms of fungal diversity. In a 30-year-fertilization experiment, Sun et al. [44] found that the carbon composition in the organic fertilizer may influence soil fungal diversity.

Other studies, including that of Luo et al. [45] and Chen et al. [18], found that organic fertilizers promoted the growth and metabolism of certain specific microorganisms, stimulating their growth and inhibiting other microorganisms' multiplication, resulting in reduced microbial diversity. In addition, it has been indicated that organic fertilizers containing microorganisms might lead to an exogenous biological invasion, representing a huge threat to biodiversity and ecosystems [46]. Summing up, the influence of organic fertilizers on the fungal diversity of soils under dry direct-seeded rice cultivation still needs intensive study.

### 4.2. Effect of Organic-Inorganic Fertilizers on Dry Direct-Seeded Paddy Soil Fungal Structure and Function

Across all treatments, Ascomycota and Basidiomycota were the most dominant phyla, whereas Sordariomycetes was the most dominant class. These findings are consistent with the results of Li et al. [47]. Nevertheless, the relative abundance of fungal phyla and classes varied under different fertilization regimes. Compared to CF, OF1 and OF3 increased the relative abundance of soil Ascomycota but decreased the relative abundance of Basidiomycota. Oppositely, OF2 and OF4 increased the relative abundance of soil Basidiomycota but decreased the relative abundance of Ascomycota. The relative abundance of Sordariomycetes decreased in OF1 and OF2 and increased in OF3 and OF4.

Fungal communities were regulated by resource type and availability [48], and although many fungal taxa may display a wide range of resource utilization capabilities, some taxa target specific organic compounds [49]. The organic fertilizers used in this study differed in their organic matter components, and this may explain, at least in part, the impact of these fertilizers on the fungal community structure. In addition, the correlation analysis demonstrated that Ascomycota and Sordariomycetes were significantly and positively correlated with rice yield. Therefore, the treatments that increased the relative abundance of these taxa also improved rice yield.

In this study, undefined saprotroph and dung saprotroph groups occupied the dominant positions in the fungal community, accounting for more than 55% in terms of relative abundance. Notably, the relative abundance of animal pathogens was greater than that of dung saprotrophs under CF treatment, while the relative abundance of wood saprotrophs was greater than that of dung saprotrophs under OF3 treatment. Thus, the application of organic fertilizers altered the ecological functional pattern of soil fungi. Compared with CF, the inclusion of organic fertilizers increased the relative abundance of dung saprotrophs, plant pathogens, fungal parasites, and leaf saprotrophs, while the relative abundance of animal pathogens showed a significantly decreasing trend. Although some studies have shown that organic fertilizers inhibited soil pathogens [25,36,50], we found a higher relative

abundance of plant pathogens under the four organic-inorganic fertilizer treatments tested, compared to CF, while the highest relative abundance of animal pathogens was detected in CF, a finding inconsistent with those of some previous studies.

Fungi have a strong nitrogen and phosphorus absorption capacity [51]; fertilizers can also nourish pathogens while promoting the growth of beneficial fungi. Organic fertilizers need to reach the stage of decomposition to allow nutrient release, and because transformation is slow, these fertilizers can provide nutrients for plants for a long time, even in the mature period; this promoting effect may also exist and account for the higher relative abundance of plant pathogens in combined organic-inorganic fertilizer treatments than conventional fertilizer alone. Furthermore, Bonanomi et al. [52] reported that the ability of organic additions to inhibit plant diseases varied greatly according to the pathogens involved. Schönning et al. [53] suggested that pathogens or harmful microorganisms in organic fertilizers may induce deleterious effects. Further research on the relationship between organic fertilizers and plant pathogens thriving in dry direct-seeded paddy soil is necessary.

## 5. Conclusions

The influence of different organic fertilizers on soil fungal community and rice yield in dry direct-seeded rice was explored by using six fertilizer treatments. Compared with those associated with CF, the Simpson and Shannon indices associated with biochar organic fertilizer treatment increased by 1.44% and 2.79%, respectively. The effect of organic fertilizers on fungal diversity depends on the type of organic fertilizer used. Ascomycota, Basidiomycota, and Sordariomycetes were dominant in the fungal community of dry direct-seeded rice; the application of organic fertilizer improved the community structure of soil-dominant bacteria; compared with the conventionally used NPK fertilizer, organic fertilizers (seaweed biological organic fertilizer + conventional NPK fertilizer, Jishiwang bioorganic fertilizer + conventional NPK fertilizer, attapulgite organic fertilizer + conventional NPK fertilizer) improved the ecological function of soil fungi, and organic fertilizers treatments increased the abundance of wood saprophytic bacteria with an average increase of 22.55–76.09%. The average reduction in animal pathogens was 28.61–73.84%. The yield of rice treated with Jishiwang bioorganic fertilizer+ conventional NPK fertilizer was the highest (15.4% higher than that obtained using the conventionally used NPK fertilizer). However, many points regarding the relationship between organic fertilizers and soil fungi remain unknown and deserve further research.

**Author Contributions:** Conceptualization, Q.G. and Y.M.; Data curation, F.S. and D.Y.; Formal analysis, J.L.; Funding acquisition, Y.L. and G.Z.; Investigation, L.X.; Project administration, G.Z.; Supervision, Y.L.; Writing—original draft, X.G.; Writing—review and editing, Y.L. All authors have read and agreed to the published version of the manuscript.

**Funding:** This research was funded by the National Key Research and Development Program of China, grant number 2017YFD0300502–6, and the Heilongjiang Bayi Agricultural University Support Program for San Heng San Zong, grant number TDJH201802.

**Institutional Review Board Statement:** Not applicable.

**Informed Consent Statement:** Not applicable.

**Data Availability Statement:** Data available from the author.

**Conflicts of Interest:** The authors declare no conflict of interest.

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
