# Peer review of "Combined Organic and Inorganic Fertilization Can Enhance Dry Direct-Seeded Rice Yield by Improving Soil Fungal Community and Structure"

_agronomy, doi:10.3390/agronomy12051213_

Round 1

Reviewer 1 Report

Dear authors, i appreciate your well written paper on "Combined organic and inorganic fertilization can enhance dry direct-seeded rice yield by improving soil fungal community and structure". The objectives and experimental procedures followed in the study are clear investigating the effects of combined organic and inorganic fertilization on microbial structure, diversity and rice yield. The paper is well organized, the topic of research is good. it's an important work, and the manuscript has been well prepared.

Minor comments

  • Please try to show the novelty of your study in the abstract, it is very necessary to increase the attractive of this study for the readers.
  • Please use different keywords than used in the title to increase the visibility of your work
  • The lines 35-38: "China’s inorganic fertilizer …… is losing sustainability", here I would like some relevant references to be cited here especially in rice under China conditions
  • The authors should improve the introduction part, most sections have sentences strung together arbitrarily without a clear structure, and transition between sentences in the introduction is confusing. Also, I strongly advice and recommended that the authors include a general sentence reviewing the role of PGPR in enhancing plant growth and development especially physiological responses, please refer to the following studies as I believe they would be useful in this statement https://doi.org/10.1186/s12870-021-02949-z https://doi.org/10.1111/ppl.13454 and the role of inoculation method https://doi.org/10.1007/s42729-021-00727-2
  • In the results section, the authors should add a state of art and need one sentence at the end of each paragraph to show to readers what happen in the whole paragraph
  • The discussion part is well interrupted and well presented

Kind Regards,

Author Response

Response to Reviewer 1 Comments Reviewer 1# Dear authors, i appreciate your well written paper on "Combined organic and inorganic fertilization can enhance dry direct-seeded rice yield by improving soil fungal community and structure". The objectives and experimental procedures followed in the study are clear investigating the effects of combined organic and inorganic fertilization on microbial structure, diversity and rice yield. The paper is well organized, the topic of research is good. it's an important work, and the manuscript has been well prepared. Minor comments • Please try to show the novelty of your study in the abstract, it is very necessary to increase the attractive of this study for the readers. Response: Thank you for this suggestion. We have revised it (L14-20). • Please use different keywords than used in the title to increase the visibility of your work Response: Thank you for this suggestion. We have modified the keywords accordingly(L35-36). • The lines 35-38: "China’s inorganic fertilizer …… is losing sustainability", here I would like some relevant references to be cited here especially in rice under China conditions Response: Thank you for this suggestion. We have rewritten this part and cited the relevant references (L47-49 ). • The authors should improve the introduction part, most sections have sentences strung together arbitrarily without a clear structure, and transition between sentences in the introduction is confusing. Also, I strongly advice and recommended that the authors include a general sentence reviewing the role of PGPR in enhancing plant growth and development especially physiological responses, please refer to the following studies as I believe they would be useful in this statement https://doi.org/10.1186/s12870-021-02949-z https://doi.org/10.1111/ppl.13454 and the role of inoculation method https://doi.org/10.1007/s42729-021-00727-2 Response: Thank you for this suggestion. We rearranged the introduction and polished the language (red marked in Introduction). • In the results section, the authors should add a state of art and need one sentence at the end of each paragraph to show to readers what happen in the whole paragraph Response: Thank you for this suggestion. We have added a concluding sentence after the corresponding paragraph (L224-226; L272-275;L292-295;L324-326;L341-342). • The discussion part is well interrupted and well presented Response: Thank you for these comments. Kind Regards,

Reviewer 2 Report

In their research, the authors raise an interesting topic of combined organic and inorganic fertilization and its effect on dry direct-seeded rice yield. This research issue is not so innovative, but the scope of the work and the comparative nature of the research, taking into account different treatments, makes the work valuable. However some remarks should be considered by the authors.

First of all, the introduction should be slightly edited. At this moment, it sounds more like a very general review. Meanwhile, in my opinion, the introduction should contain a synthetic description of specific research carried out by other authors on a given research topic. The introduction should also more precisely indicate what is new in the authors' research.

Could the authors describe in more detail the soil used in the research and its physical and chemical properties? It is quite an important element of research that describes the impact of specific fertilization variants. The same fertilization variants may have a different effect on different soil.

 I would suggest expanding the conclusions a bit by providing, for example, percentage data on increases or decreases in results.

Author Response

Response to Reviewer 2 Comments

Reviewer 2 #:

In their research, the authors raise an interesting topic of combined organic and inorganic fertilization and its effect on dry direct-seeded rice yield. This research issue is not so innovative, but the scope of the work and the comparative nature of the research, taking into account different treatments, makes the work valuable. However some remarks should be considered by the authors.

First of all, the introduction should be slightly edited. At this moment, it sounds more like a very general review. Meanwhile, in my opinion, the introduction should contain a synthetic description of specific research carried out by other authors on a given research topic. The introduction should also more precisely indicate what is new in the authors' research.

Response: Thank you for this suggestion. We have rearranged the introduction based on your suggestions and polished the language (red marked in Introduction).

Could the authors describe in more detail the soil used in the research and its physical and chemical properties? It is quite an important element of research that describes the impact of specific fertilization variants. The same fertilization variants may have a different effect on different soil.

Response: Thank you for this suggestion. We have added the details of the physiochemical properties of the soil (Table 1).

 I would suggest expanding the conclusions a bit by providing, for example, percentage data on increases or decreases in results.

Response: Thank you for this suggestion. We have revised it (L463-466; L476-479).
